# Thermo-Hydraulic Performance of Pin-Fins in Wavy and Straight Configurations

**DOI:** 10.3390/mi13060954

**Published:** 2022-06-16

**Authors:** Mohamad Ziad Saghir, Mohammad Mansur Rahman

**Affiliations:** 1Department of Mechanical and Industrial Engineering, Toronto Metropolitan University, Toronto, ON M5B2K3, Canada; 2Department of Mathematics, College of Science, Sultan Qaboos University, Al-Khod PC 123, Muscat, Oman; mansur@squ.edu.om

**Keywords:** pin-fins, wavy pin-fins channel, performance criterion, pressure drop, friction factor

## Abstract

Pin-fins configurations have been investigated recently for different engineering applications and, in particular, for a cooling turbine. In the present study, we investigated the performance of three different pin-fins configurations: pin-fins forming a wavy mini-channel, pin-fins forming a straight mini-channel, and a mini-channel without pin-fins considering water as the working fluid. The full Navier–Stokes equations and the energy equation are solved numerically using the finite element technique. Different flow rates are studied, represented by the Reynolds number in the laminar flow regime. The thermo-hydraulic performance of the three configurations is determined by examining the Nusselt number, the pressure drop, and the performance evaluation criterion. Results revealed that pin-fins forming a wavy mini-channel exhibited the highest Nusselt number, the lowest pressure drop, and the highest performance evaluation criterion. This finding is valid for any Reynolds number under investigation.

## 1. Introduction

Heat extraction is a desirable topic in engineering. Nowadays, with climate change and the introduction of electrical systems such as vehicles, amongst other applications, researchers focus on extracting and storing heat. Storing heat or energy to be used later is becoming an essential practice in engineering but is in its infancy. More sophisticated means to store a large amount of heat are under investigation. This combines engineers’ and architects’ efforts in designing a unique system capable of absorbing heat and storing it for later use. In technology, dealing with cooling small area surfaces [1] is of great interest among battery developers in using different types of vehicles. Forced flow is one of these applications through channels with varying widths, up to mini-channels of a few millimeters in width. Straight and wavy tracks are amongst the ones that were investigated recently [2]. However, the users are faced with a pressure drop, thus the need for a higher pumping mechanism. Since we are moving to a smaller scale component, this is becoming an issue. On the other hand, pin-fins have been proposed to extract heat from a surface, and this technology is not new. Forced convection in pin-fins was investigated; the uniqueness of this approach is the low-pressure drop in the system. However, to the authors’ knowledge, no studies have been written on replacing channel walls with pin-fin walls or even in a wavy form. Thus, in the present article, an attempt was made to investigate the effectiveness numerically in studying forced convection in wavy wall mini-channels fabricated of pin-fins and compare the performance against straight pin-fin wall mini-channels.

Xu et al. [3] conducted experimental measurements of heat enhancement and pressure drop on a plate with different pin-fin shapes. The shapes used in the experiment were types of round shapes, quadrangle shapes, and streamlined shapes. All forms had an identical nominal diameter. The flow was turbulent, with Reynolds numbers varying from 10,000 to 60,000. Results revealed that, as for heat enhancement, the quadrangle shape performed better than the other two sets of forms. If the pressure varied between the inlet and outlet, the friction factor varied, and thus the streamlined shapes were the best due to the lowest pressure drop. Ye et al. [4] proposed a new test section composed of pin-fins and cutback surfaces. The aim was to investigate the best cooling performance at different flow rates. The pin-fin had a cross-section of round, elliptical, and teardrop shapes having constant heights. The pin-fins were located at various positions, not forming a channel wall. They found that the addition of pin-fins reduces the film coverage in the expanded cutback at a low flow rate. The fin-pins did not show heat enhancement as expected for their particular design. Pan et al. [5] investigated the effectiveness of using pin-fin wall channels compared with solid wall mini-channels. They confirmed that pin-fin walls provide the best heat enhancement. Wang et al. [6] investigated the heat transfer characteristics of a single pin-fins mini-channel. Deng et al. [7] studied the micro pin-fins located on the hot plate. Results revealed that micro pins have a significant heat boiling enhancement.

Niranjan et al. [8] investigated the forced convection numerically on a heated plate with uniformly distributed micro pin-fins; the pin-fins form a channel. They noticed that the presence of pin-fins enhanced heat transfer by 10%. Different fin shapes were investigated by Hirasawa et al. [9], as well as Matsumoto et al. [10] and by Casano et al. [11]. Later, Ahmadian-Elmi et al. [12] continued investigating the effectiveness of pin-fins in heat extraction. Alnaimat et al. [13] used pin-fins inside a channel. 

Similarly, Saghir, and Rahman [14] conducted numerical modelling of block pins inside a channel. The channel side was large enough. The numerical results concluded that heat enhancement is evident when the track contains micro-pins. However, because micro-pins act as fins, a non-uniform temperature distribution is detected with a surge in heat extraction at each pin base. Mafeed et al. [15] investigated the importance of fin-pin height in heat extraction. Different pin-fin shapes were studied, and an optimum height was detected for some pin-fins. 

Further work was performed by Boyalakuntla et al. [16] but at the larger pin-fin size. Qu and Siu-Ho [17,18] conducted an experimental measurement of the pressure drop, friction coefficient, and Nusselt number for an array of micro pin-fins. They proposed different correlations and discussed other correlations found in the literature. The finding provided an insight into heat enhancement and the dependence of the Nusselt number on the Reynolds number. Recent work by Iasiello et al. [19] and Mauro et al. [20] focused on heat enhancement in the presence of porous media. Olabi et al. [21] focused on using nanofluid in heat extraction.

According to the current review, few studies have investigated the high importance of pin-fins, and no researchers look into the significance of pin-fin distributions. In the present study, we combined two crucial effects missing from the literature review. These are mainly pin-fins forming wavy channels and height variation effects. Thus, the uniqueness of the present work. This novel concept will be investigated compared with traditional, uniformly distributed pin-fin cases and cases with no pin-fins.

## 2. Problem Description

In the present study, an attempt was made to investigate two different pin-fins mini-channel models. A previous study by the authors [1] demonstrated that wavy channels provide a marginal heat enhancement compared with straight channels. Here we investigated two types of pin-fin mini-channels: a pin-fin forming straight mini-channel and a wavy mini-channel. For both configurations, the spacing between the pins-fins is identical, and the number of rows of pin-fins is also similar. What makes the model unique is the variable pin-fin height. Here, we investigated four pin-fin sizes. Figure 1 shows the two aluminum-fabricated configurations under investigation. The base square surface of the pin-fins is 2 mm on each side, and the pin-fin heights are 2 mm, 4 mm, 6 mm, and 8 mm, respectively. The distance between two pin-fins in the x-direction is 8.05 mm, and 7.67 mm in the y-direction. The base plate of the test section has a square shape of 37.5 mm on each side and a thickness of 3.7 mm in the z-direction. The test section has a square shape of 37.5 mm and a height of 8.7 mm. The reason for such a thick base plate is to act as a step obstacle to deflect the flow toward the pin-fins. We compared these two configurations’ performance with the case of no pin-fins to determine whether having pin-fins improves the heat enhancement.

### 2.1. Differential Equations and Boundary Conditions

Solving this problem requires three sets of differential equations. The working fluid under investigation is laminar, incompressible, and steady. Since the flow is due to the forced convection, one can neglect the gravity vector from the full Navier–Stokes equations in three dimensions. The energy equation is solved for heat transfer, in which the convective terms couple the heat transfer and the fluid flow. Third, the conduction heat transfer equation is required for the solid boundary of the setup.

The three sets of equations (Navier–Stokes, energy, and heat conduction) were solved numerically using the finite element technique with the aid of the COMSOL software, Newton, USA [17]. The differential equations are omitted for the sake of duplication but are explained in detail in reference [14]. The equations were rendered non-dimensional using the following non-dimensional parameters shown in Equation (1).
(1)$$X=\frac{x}{D},\;Y=\frac{y}{D},\;Z=\frac{z}{D},\;U=\frac{u}{u_{\mathrm{in}}},\;V=\frac{v}{u_{\mathrm{in}}},\;W=\frac{w}{u_{\mathrm{in}}},P=\frac{\mathrm{pD}}{{\mu u}_{\mathrm{in}}},\;\theta =\frac{\left(T-T_{\mathrm{in}}\right)k_{w}}{q^{\prime \prime }D}$$

In the non-dimensional parameters shown in Equation (1), D represents the characteristic length equal to 18.97 mm. It is the hydraulic diameter at the larger face of the mixing chamber facing the testing section. Since the study is conducted for different flow rates represented by the Reynolds number, the inlet velocity is u_in_. Here, k_W_ is the conductivity, μ is the viscosity and ρ is the density of the fluid. As shown in Figure 2, the red arrows indicate the applied heat flux q″ in watts per m^2^. In our analysis, we set the applied heat flux as 50,000 W/m^2^. The boundary conditions are set as follows:

At the inlet: T = T_in_ and u = u_in_, in the non-dimensional form, it becomes θ = 0 and U = 1. At the outlet: the boundary is free. At the heated bottom section, the heat flux is q = q″, in non-dimensional form; it equals 1. All external surface boundaries are assumed to be insulated for no heat losses. The solid component of the model including the pins-fins are fabricated of Aluminum. Plant and Saghir [2] conducted an experiment and demonstrated minimal heat losses. However, the heater performance with time shows some weakness in delivering the proper heat flux. From time to time, heaters are replaced for accuracy. As shown in Figure 2, the flow moves in the x-direction, and the pin’s height varies in the z-direction. The two parameters which control the thermo-hydraulic effect are the Reynolds number defined as Re = $\frac{{\rho u}_{\mathrm{in}}D}{\mu }$ and the Prandtl number defined as Pr = $\frac{c_{p}\mu }{k_{w}}$.

In the present study, we investigated the role of different model parameters on the flow and thermal fields. We calculated the temperature distribution at the pin’s base where a fluid circulates from the inlet to the outlet. Then, we calculated the Nusselt number at the exact locations where the temperature is measured. The pressure drop is calculated between the inlet and the outlet, along with the friction coefficient for different configurations. Finally, the influence of the Nusselt number and the pressure drop are combined by defining the performance evaluation criterion (PEC). By definition, Nusselt number, Nu = $\frac{\mathrm{hD}}{k_{w}}$ where the heat convection coefficient, h = $\frac{q^{\prime \prime }}{\left(T-T_{\mathrm{in}}\right)}$. In dimensionless form, it becomes Nu = $\frac{1}{\theta }$. The friction factor in the non-dimensional form is defined as f=0.2529× ΔPRe, where P is the pressure and Re is the Reynolds number. Therefore, the performance evaluation criterion in non-dimensional form is defined as $\mathrm{PEC}=\frac{{\mathrm{Nu}}_{\mathrm{average}}}{f^{1/3}}$. These parameters can guide the reader on heat removal effectiveness for various configurations at different conditions. The readers are invited to read reference [14] for the detailed equations used in our model.

### 2.2. Mesh Sensitivity and Convergence Criteria

A mesh analysis was conducted to ensure the mesh used provides accurate results. Table 1 presents different mesh levels used and the corresponding Nusselt number. As can be seen, using of a normal mesh consisting over 700,000 elements provides accurate results.

In using COMSOL software, we used the default solver segregated method. The reader can refer to reference [22] for more details. In a nutshell, the convergence criteria were set as follows: at every iteration, the average relative error for U, V, W, P, and θ were computed using the following relation:(2)$$R_{c}=\frac{1}{n\;\cdot \;m}\overset{i=m}{\underset{i=1}{\sum }}\overset{j=n}{\underset{j=1}{\sum }}\left|\frac{\left(F_{i,j}^{s+1}-F_{i,j}^{s}\right)}{F_{i,j}^{s+1}}\right|<1.0\times 10^{-6}$$
where F represents one of the unknowns, viz. the velocities, pressure, and temperature, s is the iteration number, and (i, j) represents the coordinates on the grid.

## 3. Results and Discussion

Investigating the effectiveness of heat removal in the presence of a wavy and a straight mini-channel composed of pin-fins is the objective of this current study. Different parameters relating to the flow and thermal fields are examined.

### 3.1. Thermo-Hydraulic Performance of Different Pin Channels Configuration

The heat enhancement and pressure drop is investigated for various Reynolds number ranges and pin-fin heights. Figure 3 presents the temperature variation near the base of the pin-fins for different configurations and Reynolds numbers. It is observed that regardless of the design under investigation, the temperature profiles are similar, having different intensities. A maximum temperature exists near the end of the flow path for all cases in this figure.

Obviously, as the Reynolds number increases, the heat extraction increases accordingly, resulting in a temperature drop. It is also evident that the development of the flow and thermal boundary layers leads to obstruction in heat removal near the end of the flow path. Interestingly, the temperature variation at the base of the pin-fins is wavy. In particular, it is lower at the pin-fins’ bottom. The reason is that pin-fins absorb more heat than the base plate, resulting in waviness. One way to observe this finding is to investigate the average Nusselt number for all cases, as depicted in Figure 4.

By examining Figure 4, and since the average Nusselt number is known as the inverse of the temperature, the wavy pin-fins configuration confirms the previous finding. It is proven here that the pin-fins forming wavy mini-channels exhibit the highest Nusselt number and outperform the case with no pin-fins. The variation in the pin-fins’ average Nusselt number with Reynolds number for all heights is not identical. By comparing the two configurations of pin-fins arrangement, the average Nusselt number for fin-pins forming wavy mini-channels case is higher.

As mentioned earlier, the advantage of using pin-fins in mini-channels results in lower pressure drop. The average Nusselt number measuring the performance evaluation criterion for the two pin-fins configurations is depicted in Figure 5. As expected, pin-fins forming wavy mini-channels confirm the earlier conclusion, providing better heat enhancement without additional pumping power. The larger pin-fin sizes demonstrate a more vital performance evaluation criterion for both pin-fins configurations.

Finally, Figure 6 presents the hydrodynamic effect by displaying the flow behavior in different planes when the pin-fin height is 6 mm and the Reynolds number equal to 250. In Figure 6, column A is for straight pin configuration and column B for Wavy pin configuration. Comparing Figure 6a,d it is evident that the flow circulation in the latter one is more effective in removing heat from the bottom of the plate. Flow circulation is more pronounced compared with Figure 6a. What triggered this flow is shown in Figure 6b,e. A step obstacle at the bottom of the mixing chamber forces the flow to jump and penetrate the pin-fins, creating a mixture that remains in the laminar regime. We again re-examined Figure 6a,d by displaying the flow patterns in Figure 6c,f, respectively. Indeed, a more complex flow circulation is evident in Figure 6f; thus, the reason for the better performance evaluation criterion. In Figure 6c, the flow moves in mini-channels; whereas in the wavy shape Figure 6f, it circulates non-uniformly, resulting in better heat extraction. The flow circulates between the pin-fins and the top of the pin-fins, as the pin-fins’ height is smaller than the cavity depth. Thus, one can see a counter flow emerging from the bottom of the hot plate to the top of the test section. This helps increase the mixing and heat extraction.

### 3.2. Heat Removal for All Configurations

It is interesting to calculate the amount of heat removed from the system for all configurations. The heat removal is known to be Q = ${\;\dot{m}c}_{p}\left(T_{\mathrm{out}}-T_{\mathrm{in}}\right)$, where $c_{p}$ is the specific heat, T_in_ and T_out_ are the temperatures at the flow inlet and flow outlet, respectively.

The $\;\dot{m}$ is the mass flow rate which is the product of the flow rate and the density of the water. In the non-dimensional form, the formulation becomes Re.Pr.$\theta_{\mathrm{out}}$. The inlet temperature is related to the definition of the non-dimensional temperature presented in Equation (1). Figure 7 illustrates the three configurations’ heat removed for different pin-fins’ heights. The first observation, which applies to all the cases, is that the absence of pin-fins is the worst case for heat removal. This observation is followed by the fact that pin-fins forming a wavy mini-channel has better heat removal than the straight channel. This confirms all previous findings by showing the temperature, the Nusselt number, or the PEC. For all cases, the non-linear behavior of the heat removal profile suggests that at the Reynolds number of 150, less heat is removed from the system. However, what is evident is that the best heat removal mechanism is when the pin-fins’ height is 4 mm. That means flow going through the pin-fins and above the pin-fins plays a significant role in heat extraction.

### 3.3. Friction Coefficient and Nusselt Number Correlations

It is worth discussing the relation between the friction factor, the Nusselt number, and the Reynolds number. Qu and Siu-Ho experimented with an array of pin-fins staggered along with the flow and proposed a relation between the friction factor and the Reynolds number. Their experimental setup and the pin-fins distributions are not similar to our proposed model. Figure 8 summarizes our numerical findings of the variation in the friction coefficient with the Reynolds number.

In addition, in the same plot, the experimental findings of Qu and Siu-Ho [17,18] are plotted. As one can observe, the trend for the friction factor is identical, and the friction factor values are also within the obtained values. This indicates that the numerical results obtained are reasonable and correct. As shown in Figure 8, an increased Reynolds number reduces the friction coefficient accordingly. For the case of pin-fins forming a wavy channel, as indicated earlier, greater friction is expected than the pin-fins creating straight channels. A fitting curve for Figure 8 (not including Qu and Siu-Ho curves) is summarized by two analytical relations shown in Equations (3) and (4) as a function of Reynolds number and the pin-fins’ height. The friction factor trend variation with the Reynolds number is well-presented in these two equations having a 97% accuracy.

For straight mini-channels:(3)f=59.355Re−0.733HD0.488

For wavy mini-channels:(4)f=38.49Re−0.707HD0.353

Furthermore, a correlation between the average Nusselt number as a function of the Reynolds number and the pin-fins’ height was determined for the two cases of pin-fins distribution in the mini-channels. Equations (5) and (6) display the obtained relations for different Reynolds numbers and heights. It is essential to indicate that these relations are valid for a range of Reynolds numbers from 0 to 250 and for a range of pin-fins size from 2 mm to 8 mm with an accuracy of 97%.

For straight mini-channels:(5)Nuaverage=2.848Re0.398HD0.358

For wavy mini-channels:(6)Nuaverage=2.82Re0.402HD0.359

It is evident from these two relations that the pin-fins forming wavy channels exhibit a higher Nusselt number.

## 4. Conclusions

In the present study, we investigated the thermo-hydraulic performance of a set of pin-fins forming wavy and straight mini-channels. For this purpose, the Navier–Stokes equations and the energy equations were solved numerically using the finite element technique. Different variables were examined in this study, mainly the variation in Reynolds number and the height of the pin-fins. Thus, two flows were involved in the test section, primarily the first flow circulating between the pin-fins and the other one spreading above the pin-fins. This setup’s thermal and hydraulic performance was compared with the case of pin-fins forming straight mini-channels and having no pin-fins. Results revealed;

The pin-fins forming wavy mini-channels exhibit more significant heat enhancement and less pressure drop;The performance evaluation criterion is found to be higher for the wavy mini-channel configuration compared with the straight pin-fins configuration;The flow above the pin-fins is found to reduce the pressure drop.

## Figures and Tables

**Figure 1 micromachines-13-00954-f001:**
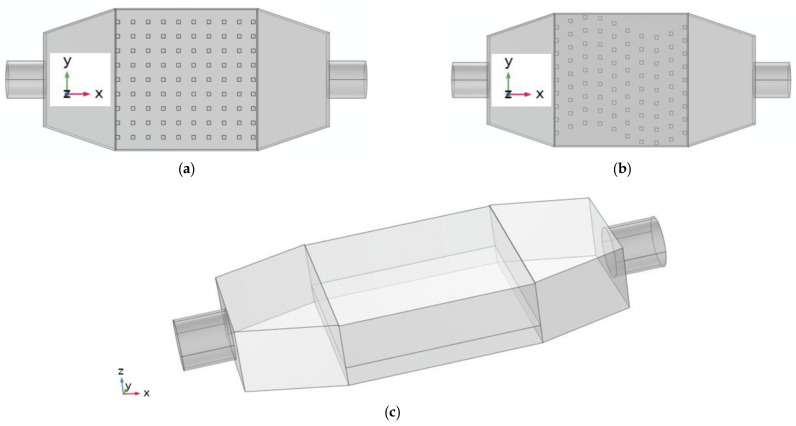
Three different configurations ((**a**) Straight pin-fins configuration, (**b**) Wavy pin-fins configuration, (**c**) No pin-fins configuration).

**Figure 2 micromachines-13-00954-f002:**
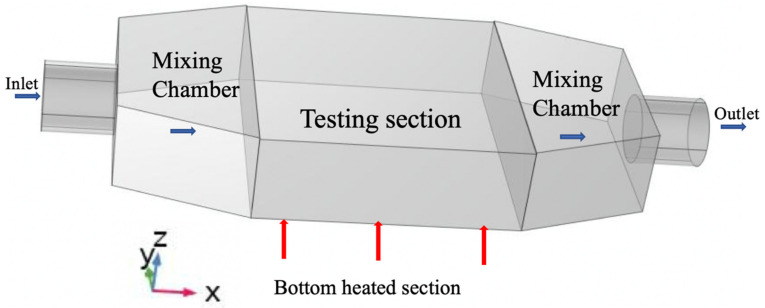
Boundary conditions.

**Figure 3 micromachines-13-00954-f003:**
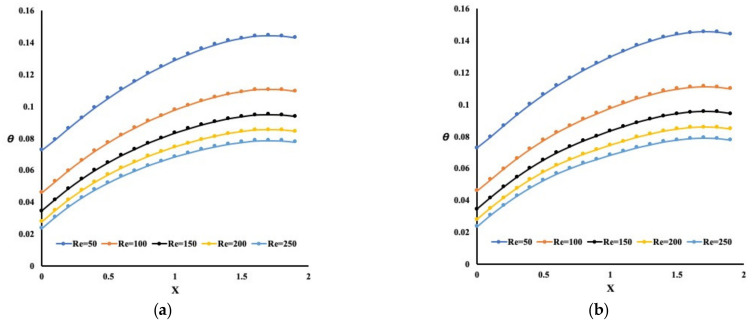
Temperature variation for different configurations (6 mm pin-fins height case, (**a**) Straight pin-fins configuration, (**b**) Wavy pin-fins configuration, (**c**) No pin-fins configuration).

**Figure 4 micromachines-13-00954-f004:**
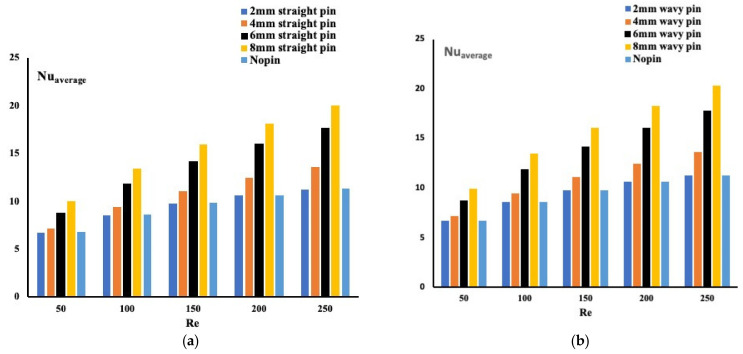
Average Nusselt number for different configuration ((**a**) Straight pin-fins configuration, (**b**) Wavy pin-fins configuration).

**Figure 5 micromachines-13-00954-f005:**
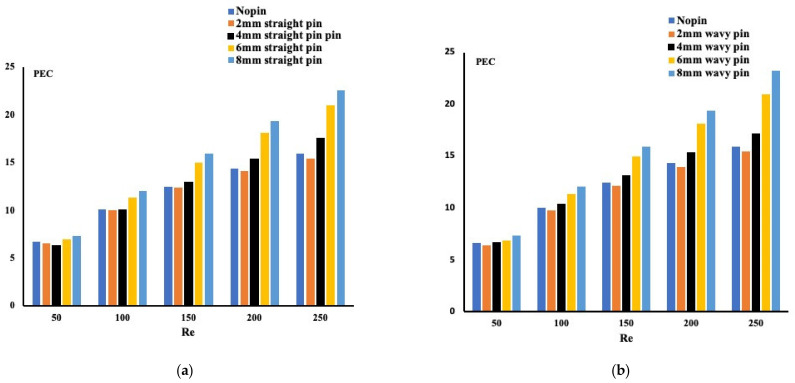
Performance evaluation criterion for different configurations ((**a**) Straight pin-fins configuration, (**b**) Wavy pin-fins configuration).

**Figure 6 micromachines-13-00954-f006:**
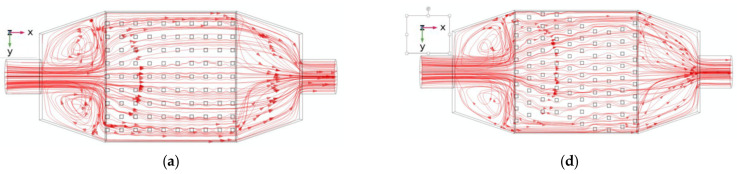
Flow distributions for the pin configurations. (Re = 250, pin-fin height 6 mm, (**a**) Streamline in (xy) plane, (**b**) Streamline in (xz) plane, (**c**) Velocity in (xy) plane, (**d**) Streamline in (xy) plane, (**e**) Streamline in (xz) plane, (**f**) velocity in (xy) plane).

**Figure 7 micromachines-13-00954-f007:**
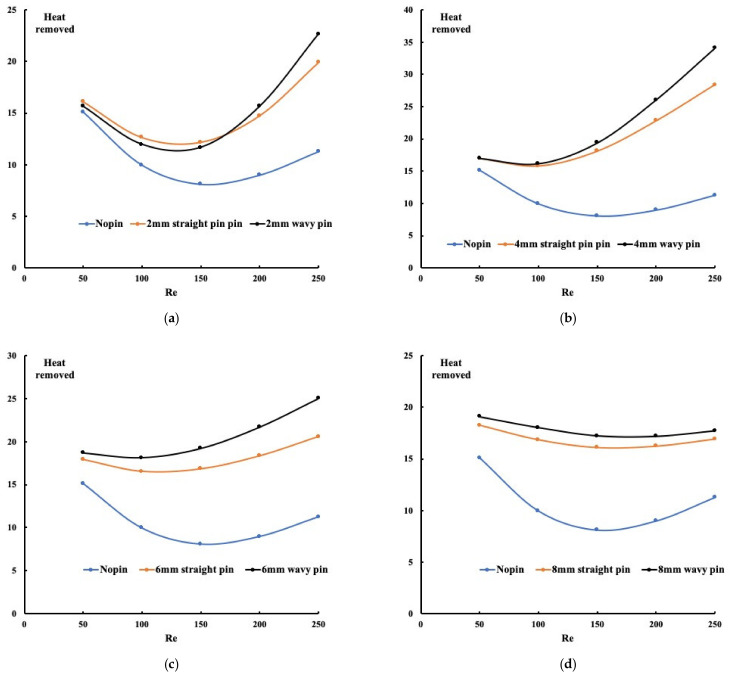
Heat removed for all configurations ((**a**) 2 mm pin height, (**b**) 4 mm pin height, (**c**) 6 mm pin height, (**d**) 8 mm pin height).

**Figure 8 micromachines-13-00954-f008:**
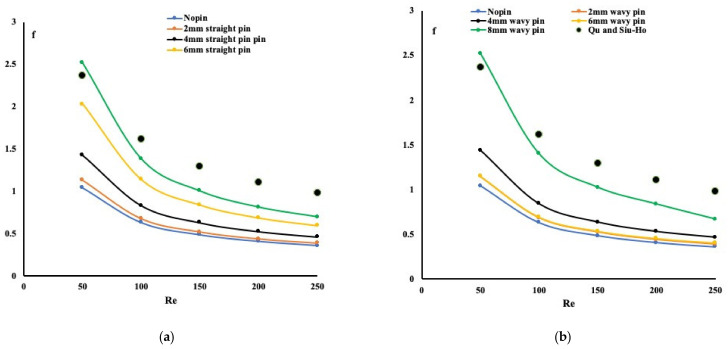
Variation in friction coefficient with Reynolds number ((**a**) Straight pin-fins mini-channels, (**b**) Wavy pin-fins mini-channels).

**Table 1 micromachines-13-00954-t001:** Mesh sensitivity analysis.

Mesh Name	Number of Elements	Average Nusselt Number
Normal	700,494	7.157
Coarse	599,637	7.153
Coarser	301,134	7.106

## Data Availability

Not applicable.

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
