# Peer review of "Thermo-Hydraulic Performance of Pin-Fins in Wavy and Straight Configurations"

_micromachines, 2022, doi:10.3390/mi13060954_

Round 1
Reviewer 1 Report
1. The paper is titled of “Thermo-hydraulic performance of pin-fins in wavy and straight channels configurations”. But according to the problem description of the present work, the words of “wavy and straight” refer to the geometric features of the pin-fins’ configuration, not that of the channel wall? So the title is confused, should be corrected.
2. What application background is based to establish the geometric models described in Figure 1? Why a circular inlet and outlet, a front and a rear mixing chamber are included? Figure 1 should be revised to give more geometric information including the sizes of pin-fins and the channels, the distribution of the pin-fins to help authors’ understanding. What is the heights of the channels (the test section and the mixing chambers)? What’s the meaning of that “The reason for such a thick base plate is to act as a step obstacle to deflect the flow toward the pin-fins.”?
3. The used boundary conditions are not described clearly. How the conjugated heat transfer between the fluid and solid region (including pin-fins and the thick base plate) is realized? What are the thermo-physical properties of the solid region?
4. In the section of “2.1. Differential equations and boundary conditions”, formulas and their symbols’ description are all placed in the text, which is very confusing, and not all symbols are explained, such like how the temperature T is defined in θ.
5. In most of the figures, why is the physical quantity of y axis placed on the right of the y axis? except Fig.3. The quality of the figures are poor.
6. How the average Nusselt number is defined? Is it possible that the average Nusselt number is equal to each other for the cases of nofin and 2mm straight fin and wavy fin? In the other words, if they are equal, it means pin-fins with height of 2mm do not affect the heat transfer of the channel, regardless of the pin-fins’ distribution, why?
7. Why the results of average Nusselt number for the cases of nofin, straight fin and wavy fin are not compared in one figure?
8. Fig. 7 shows that at Reynolds number of 150 or 100, heat removal capacity is lowest, but according to Fig. 5, average Nusselt number increases with the increase of Reynolds number, why the average Nusselt number cannot represent the heat removal capacity?
9. In the introduction of the paper, it is mentioned in several places that the no or few paper carried out the effect of pin-fins’ distribution and height on the channel flow and heat transfer, such as “… to the authors’ knowledge, no papers have been found in replacing channel walls with pin-fin walls…”, “…According to the current review, few studies investigate the high importance of pin fins, and no researchers look into the significance of pin-fin distributions…”. It is not true because that the effect of pin-fins’ distribution and height on the channel flow and heat transfer have received much attention for the real application of pin-fins heat transfer enhancement. The authors should review more literature to understand that.
10. The writing and typesetting of paper is not good, there are many errors, such as:
in the abstract, “The thermo-hydraulic performance of the three configurations is determined studied the Nusselt number…”, what the meaning of “determined studied”?
in the introduction, “Since we are moving to nano-science and nano-technology, it is becoming an issue.”, but the present work is not about nano-science and nano-technology.
In the conclusion, “Thus, two flows were involved in the test section, primarily the first flow circulating between the pin-fins and the other one spreading above the pin-fins.”, but the present paper did not present any detailed flow field of the second flow spreading above the pin-fins.
Author Response
Thank you for taking the time to review our paper. You have raised some very innovative and vital points; I am grateful for them. I hope my correction meets your expectation. I have highlighted the edits in yellow for clarity.

Reviewer 2 Report
In this contribution, the authors compare some different heat exchanger to be employed for turbines. These exchangers are equipped with pin-fins and wavy pin-fins, forming minichannel in which air as the coolant fluid passes. Results underline that wavy pin-fins present the best performances in terms of Nusselt number, as well as lowest pressure drop and highest performance evaluation criterion. The reviewer think that this paper present some interesting results about improving the performances of heat exchangers; therefore it can be considered for publication if the authors answer to the following points
· - In the introduction, the authors present the current state-of-art about techniques currently used to improve heat transfer like finned surfaces. Presenting the current state-of-art is very important to appreciate the novelty from a paper, and this is something that the authors carefully did. However, in order to underline that among various solutions using wavy pin-fins could be of interest, they forgot to mention other relevant solutions for heat exchangers like metal foams [1, 2] or nanofluids [3]. The authors should mention all this in the introduction in order to appreciate the novelty from the present study
· - How did the authors decide the wavy pin-fin configuration from a mathematical point of view? Are they assuming the same volume of fins, correct?
· - If available, did the authors have the chance to compare their results with similar experimental or numerical studies from the literature?
· - The authors should report governing equations here employed, together with an exhaustive description of the assumptions here done. For instance, did the authors check that a laminar flow assumption is here reasonable?
· - The authors assume here that the applied heat flux is 50,000 W/m2. Is this assumption reliable? Why didn't they consider a uniform temperature? Besides, did the authors consider temperature-related effects like thermophysical properties variation, or radiation?
· - Some more details about numerical modeling is suggested; for instance, which solver (direct? iterative?) did the authors use to solve governing equations?
· - Why all the curves in Fig. 3 present a local maximum for X equal to about 1.75? Besides, these curves seem to be parameteric with respect to Reynolds number. Did the authors try to scale their results in order to obtain just one curve for all the cases investigated?
· - The authors present different performance evaluation criterion in Fig. 5. How did the authors compute these index? Are they assuming that the different heat exchangers are compared at equal surface area and pumping power (see [4] for a resume of all the different definitions of performance evaluation criteria done through the years)
· - In Fig. 6d, the authors show streamlines in xy plane. Did they observe some recirculation at the top-right of the domain, which might happen because the flow faces a large free area? Similar observations might be done for the bottom-left part of the investigated domain
· - As a suggestion, why didn't the authors consider a wavy-pin distribution with variable spacing between pin-fins? This could be an interesting solution to optimize heat transfer here
· - Why in Fig. 7 the authors found a local minimum in all the figures, which depends on the configuration investigated? Is this minimum somehow related to the dimensionless temperature profiles introduced in Fig. 3?
· - Please provide some statistical parameters like coefficient of determination for correlations depicted in Eqs. (3)-(6)
· - Please use bullet points in the conclusions, together with some quantitative results that arise pro/cons of the configurations here investigated
[1] Iasiello, M., Bianco, N., Chiu, W. K. S., & Naso, V. (2021). The effects of variable porosity and cell size on the thermal performance of functionally-graded foams. International Journal of Thermal Sciences, 160, 106696.
[2] Mauro, G. M., Iasiello, M., Bianco, N., Chiu, W. K. S., & Naso, V. (2022). Mono-and Multi-Objective CFD Optimization of Graded Foam-Filled Channels. Materials, 15(3), 968.
[3] Olabi, A. G., Wilberforce, T., Sayed, E. T., Elsaid, K., Rahman, S. A., & Abdelkareem, M. A. (2021). Geometrical effect coupled with nanofluid on heat transfer enhancement in heat exchangers. International Journal of Thermofluids, 10, 100072.
[4] Yilmaz, M., Comakli, O., Yapici, S., & Sara, O. N. (2005). Performance evaluation criteria for heat exchangers based on first law analysis. Journal of Enhanced Heat Transfer, 12(2).
Author Response
Thank you for taking the time to review our paper. You have raised some very innovative and vital points, for which I am grateful. I hope my correction meets your expectation. I have highlighted the revisions in blue for clarity. Below are our answers to your questions.

Round 2
Reviewer 2 Report
The paper can be accepted as it is in the revised version
Author Response
Dear Colleague
Thank you for accepting the rebuttal
best regards
Ziad
